Exploring the relationship between nutritional intake and menstrual cycle in elite female athletes

Miyamoto Mana 1 2 mana-miyamoto@nuhw.ac.jp
http://orcid.org/0000-0002-8252-6174 Shibuya Kenichi 1 2
1 Department of Health and Nutrition, Niigata University of Health and Welfare , Niigata , Japan
2 Graduate School of Health and Welfare, Niigata University of Health and Welfare , Niigata , Japan
Kabir Russell
Electronic publication date: 2023 Sep 25
Publication date: 2023
Volume: 11
Electronic Location ID: e16108
Received 2023 Jun 20; Accepted 2023 Aug 25
Copyright: © 2023 Miyamoto and Shibuya
Copyright year: 2023
Copyright holder: Miyamoto and Shibuya
License: This is an open access article distributed under the terms of the Creative Commons Attribution License, which permits unrestricted use, distribution, reproduction and adaptation in any medium and for any purpose provided that it is properly attributed. For attribution, the original author(s), title, publication source (PeerJ) and either DOI or URL of the article must be cited.
License URL: https://creativecommons.org/licenses/by/4.0/

Keywords: Elite athletes, Cortisol, Estrogen, Progesterone, Menstrual cycle

Funding: The authors received no funding for this work.

==============================
This study aimed to examine potential variations in nutritional intake among female athletes, including top-level, throughout the menstrual cycle. A total 122 female athletes participated in the study, documenting their food consumption over a 3-day period. The menstrual status of female athletes was also assessed, and using the survey results, the phase of the menstrual cycle (the follicular, early luteal, or late luteal) during which each meal was recorded was determined. Consequently, the meal records were categorized into the respective three phases. The findings of this study indicated that there were no notable disparities in nutritional intake, encompassing energy, protein, fat, carbohydrate, calcium, iron, and fiber, across the three phases of the menstrual cycle. The results imply that female athletes may experience comparatively smaller variations in nutrient intake related to the menstrual cycle. This could be attributed to the higher energy requirements of female athletes because of their rigorous training. This study underscores the significance of accounting for the population when examining nutrient intake changes associated with the menstrual cycle.

Introduction

Preserving optimal menstrual cycle function plays a vital role in safeguarding female reproductive health (Rogan & Black, 2022). The menstrual cycle encompasses a multifaceted sequence of events that primes the uterus for potential pregnancy. It is categorized into two primary phases, namely the follicular and luteal phases, distinguished by the onset of menstruation, follicular maturation, ovulation, and the formation of the corpus luteum (Reed & Carr, 2000).

Nevertheless, solely dividing the menstrual cycle into these two phases does not offer sufficient insight to comprehensively grasp the diverse hormonal fluctuations that transpire within them. Consequently, researchers frequently employ sub-phases, such as follicular, ovulatory, early luteal, and late luteal phases, to categorize the menstrual cycle with greater precision (Hirschberg, 2012). These sub-phases are characterized by notable shifts in hormone levels, including estrogen, progesterone, follicle-stimulating hormone (FSH), and luteinizing hormone (LH) (McLay et al., 2007). The menstrual cycle commences with the follicular phase, triggered by the onset of menstruation. During this initial phase, ovarian follicles gradually mature in response to follicle-stimulating hormone (FSH) released by the pituitary gland. As these follicles develop, the production of estrogen increases. This estrogen surge exerts a profound influence on diverse bodily systems, encompassing metabolism, thermoregulation, and neurotransmission. Furthermore, heightened estrogen levels prompt the pituitary gland to release luteinizing hormone (LH), a pivotal trigger for ovulation. Subsequent to ovulation, in the early luteal phase, the secretion of progesterone commences, accompanied by sustained estrogen production. Simultaneously, the secretion of FSH and LH is repressed. As the menstrual cycle progresses into the late luteal phase, the corpus luteum, once vibrant, initiates regression. Consequently, levels of both estrogen and progesterone gradually wane.

Moreover, it is widely recognized that both estrogen and progesterone influence feeding behavior. Estrogen is thought to have appetite-suppressing effects, whereas progesterone has been demonstrated to increase appetite, particularly when combined with estrogen (Devries et al., 2006; Elliott-Sale et al., 2021; Giersch et al., 2020). Research conducted on the non-athlete individuals in the general population has indicated a noteworthy reduction in energy intake during the follicular phase of the menstrual cycle compared to the luteal phase. These variations are attributed to the hormonal effects observed during this period (Elliott-Sale et al., 2021; Giersch et al., 2020; Kammoun et al., 2017; Barr, Janelle & Prior, 1995; Cheikh Ismail et al., 2009). To clarify, during the follicular phase, we anticipate a decrease in appetite attributed to estrogen’s influence. Conversely, in the early luteal phase, we expect a notable increase in appetite due to the combined effects of estrogen and progesterone. However, during the late luteal phase, while the impact of these hormones is somewhat diminished, the presence of menstrual cramps might further dampen the appetite. Indeed, prior investigations have unveiled a noteworthy rise in nutrient consumption during the initial half of the luteal phase when contrasted with the follicular phase within the broader female population (Bowen & Grunberg, 1990; Tucci et al., 2009; Martini et al., 1994; Tarasuk & Beaton, 1991).

However, fluctuations in energy intake during the menstrual cycle can exert a considerable influence on the sports performance of female athletes. Hence, it is imperative to examine the variability of energy intake across the menstrual cycle within this population. It is widely recognized that the menstrual cycle can impact athletic performance (Freemas et al., 2021; Lebrun et al., 1995) as well as exercise metabolism (Oosthuyse & Bosch, 2010). Moreover, it is well-established insufficient energy intake, considering the elevated energy expenditure of female athletes, could lead to menstrual and other physiological dysfunction (Miyamoto, Hanatani & Shibuya, 2021a). However, despite the unique challenges faced by female athletes in this aspect compared to their male counterparts, there is a dearth of research in sports science that specifically focuses on female athletes (Carmichael et al., 2021; Cowley et al., 2021). Consequently, it remains unclear whether the impact of sex hormones on energy intake is significant among female athletes.

The objective of this study was to examine variances in nutrient intake throughout the menstrual cycle among female elite athletes. Considering the augmented energy expenditure associated with physical exercise, top-level athletes possess higher greater energy demands compared to the general population. Therefore, we hypothesized that there would be no significant difference in nutrient intake during the menstrual cycle. To assess the validity of this hypothesis, the current study scrutinized the dietary intake and menstrual cycles of 122 female athletes, encompassing individuals competing at the international level. These data were subjected to analysis through a generalized linear mixed model. This study would provide fundamental data for developing measures to maintain the condition of female athletes.

Methods

Participants

The present study recruited 122 female athletes between the ages of 15 and 24 years, who were intercollege-level track and field athletes and international-level rowing athletes. All athletes included in the study provided their consent to participate. The study was approved by the Ethics Committee of Niigata University of Health and Welfare (Approval #17982-180606), and each participant provided written informed consent after receiving a comprehensive explanation of the study procedures and the non-invasive nature of the research.

Dietary intake, body mass, and body composition

The participants utilized a meal-recording method to document their dietary intake. Every athlete independently selects 3 days per week, comprising two training days and one designated rest day. They captured photos of each meal, including breakfast, lunch, dinner, and snacks, which were then sent to the team dietitian. The dietitians analyzed the nutrient content of these meals using information obtained from the Japan National Nutrient Database or specific nutrition facts panels provided by the products. Daily nutrient intake data, such as total energy (kcal), macronutrients (grams and percentage), and fiber (in grams), were averaged. Participant measurements of body mass (BM) and body fat percentage (%BF) were taken using a commercially available home scale (BC-314; Tanita Co., Tokyo, Japan). Body mass index (BMI) was calculated by dividing weight (in kilograms) by height squared (in meters). Height, BM, and %BF were measured only once during the investigation. All these methods were consistent with those used in previous studies (Miyamoto, Hanatani & Shibuya, 2021a, 2021b; 2022).

Menstrual cycle

The participants documented their menstrual cycles by utilizing a paper and pen calendar and provided the information through reporting. Using the data of the last menstrual period and the length of each individual’s menstrual cycle, we determined the phase of the menstrual cycle to which the meal survey period corresponded. In cases where a clear determination could not be made, or if an athlete had an irregular menstrual cycle, the data were excluded from the analysis.

Statistical analyses

The data were expressed as mean ± standard error. We conducted statistical analyses using the lmerTest package in R (version 4.2.3) (R Core Team, 2023). We used Akaike’s Information Criterion (AIC) to validate the parameters for the linear mixed model (LMM). For the LMM analysis, subject ID was set as a random effect, and fixed effects were identified based on the AIC score, which served as an indicator. Then, we performed the analysis of variance using the results of this LLM, and we estimated the degrees of freedom using the Kenward-Roger method. The significance level was set as 0.05.

Results

Physical characteristics

Table 1 shows the physical characteristics of athletes. The average body fat percentage (%BF) among the athletes was 21.7%, with a 95% confidence interval (CI) of [20.7–22.7%]. Out of the 122 athletes, 45 had a %BF below 20%. The average body mass index (BMI) was 21.9, with a 95% CI of [21.7–22.2]. None of the athletes had a BMI below 18.5, but 17 had a BMI below 20.

Table 1 Mean values of physical characteristics and state anxiety (Mean ± S.E).

Height (cm)	Body mass (kg)	%BF (%)	BMI	
168.0 ± 0.4	62.0 ± 0.5	21.7 ± 0.5	21.9 ± 0.1	
Note:

BF, body fat; BMI, body mass index.

Macronutrient intake

The mean energy intake of all athletes was 2,626.1 kcal (95% CI [2,532.6–2,719.7 kcal]) (Fig. 1) During the dietary survey period, 61 of 122 athletes were in the follicular phase, 40 were in the early luteal phase, and 21 were in the late luteal phase. The mean energy intake for these phases was 2,732.6 kcal (95% CI [2,601.5–2,863.8 kcal]), 2,492.7 kcal (95% CI [2,322.6–2,662.8 kcal]), and 2,571.1kcal (95% CI [2,379.3–2,762.9 kcal]), respectively (Table 2).

Figure 1 Changes in nutrient intake per menstrual cycle.

There were no statistically significant differences in nutrient intake between menstrual cycles (p > 0.05).

Table 2 Nutritional intake during the menstrual cycle.

Explanatory variables	Total	Follicular	Luteal early	Luteal late	F value	p value	
95% CI	n = 122	n = 61	n = 40	n = 21			
Energy (kcal/day)	2,626.1	2,732.6	2,492.7	2,571.1	1.574	0.212	
95% CI	[2,532.6–2,719.7]	[2,601.5–2,863.8]	[2,322.6–2,662.8]	[2,379.3–2,762.9]			
CHO (g/day)	343.8	360.6	319.2	342.3	1.886	0.157	
95% CI	[328.7–359.0]	[338.2–382.9]	[294.5–343.8]	[308.8–375.7]			
Protein (g/day)	112.0	116.3	107.2	108.7	1.213	0.301	
95% CI	[107.3–116.7]	[109.6–122.9]	[98.6–115.7]	[99.3–118.2]			
Fat (g/day)	84.9	87.1	82.9	82.6	0.562	0.572	
95% CI	[81.2–88.7]	[81.6–92.5]	[76.1–89.7]	[75.2–89.9]			
Ca (mg/day)	831.8	885.9	776.8	799.6	1.007	0.369	
95% CI	[767.3–896.4]	[779.1–992.7]	[689.0–864.7]	[656.3–902.8]			
Fe (mg/day)	12.5	13.2	11.7	12	1.322	0.271	
95% CI	[11.8–13.2]	[12.2–14.1]	[10.4–13.0]	[10.7–13.4]			
Fiber (g/day)	16.6	17.2	15.6	16.3	0.734	0.497	
95% CI	[15.6–17.5]	[15.7–18.7]	[14.1–17.2]	[14.4–18.1]			
Vitamin B1 (mg/day)	1.77	1.8	1.67	1.85	0.724	0.487	
95% CI	[1.37–1.87]	[1.66–1.94]	[1.51–1.84]	[1.57–2.13]			
Vitamin B2 (mg/day)	2.13	2.17	2.07	2.12	0.004	0.962	
95% CI	[1.98–2.27]	[0.95–2.38]	[1.83–2.30]	[1.74–2.51]			
Vitamin C (mg/day)	181.3	178.5	179.9	191.9	0.498	0.609	
95% CI	[164.6–197.9]	[154.7–202.3]	[151.2–208.5]	[149.9–233.9]			
Note:

Total: Encompassing the entire dataset, this category includes all 122 athletes who were part of the study. Follicular: This category includes 61 athletes whose study period falls within the follicular phase of the menstrual cycle. Luteal Early: The category includes 40 athletes whose study period falls within the initial half of the luteal phase of the menstrual cycle. Luteal Late: The category includes 21 athletes whose study period falls within the latter half of the luteal phase of the menstrual cycle.

The mean protein intake was 112.0 g (95% CI [107.3–116.7 g]). The mean protein intake per body mass (Pro/BM) was 1.8 g/kg (95% CI [1.7–1.9 g/kg]). Among the athletes, Pro/BM values of 1.0 or above, 1.5 or above, and 2.0 or above were found in 122, 88, and 43 athletes, respectively. The mean carbohydrate (CHO) intake was 343.8 g (95% CI [328.7–359.0 g]). The mean CHO intake per body mass (CHO/BM) was 5.6 g/kg (95% CI [5.3–5.8 g/kg]). CHO/BM values of five or above, and six or above were found in 70 and 44 athletes, respectively.

Micronutrient intake

The mean calcium (Ca) intake was 831.8 mg (95% CI [767.3–896.4 mg]) (Table 2). Among the athletes, 46 athletes had a Ca intake of more than 900 mg, and 18 athletes had a Ca intake of more than 1,200 mg. The mean iron (Fe) intake was 12.5 mg (95% CI [11.8–13.2 mg]). Among the athletes, 34 athletes had a Fe intake below 10 mg, 94 athletes had a Fe intake below 15 mg, and 110 athletes had a Fe intake below 18 mg.

Effect of the menstrual cycle

Table 2 shows that there was no significant difference in energy intake between the follicular, early luteal, and late luteal phases (F = 1.573, p = 0.212). Additionally, there were no significant differences in protein, fat, or carbohydrate intakes between these phases (F = 1.213, p = 0.301; F= 0.562, p = 0.572; F = 1.886, p = 0.157, respectively). There were also no significant differences in Ca, Fe, or fiber intakes between these phases (F = 1.007, p = 0.369; F = 1.322, p = 0.271; F = 0.734, p = 0.497, respectively).

Discussion

This study stands as a pioneering endeavor in elucidating the intricate interplay between the menstrual cycle and nutritional intake among female athletes, encompassing those of elite caliber. As such, we consider this contribution to hold substantial significance in the field. The present study revealed no significant variation in nutritional intake among female elite athletes throughout the menstrual cycle. This finding diverges from numerous studies conducted on the general female population, which have consistently demonstrated higher nutrient intake during the luteal phase compared to the follicular phase (Bowen & Grunberg, 1990; Tucci et al., 2009; Martini et al., 1994; Tarasuk & Beaton, 1991). Moreover, feeding behavior is known to be influenced by both estrogen and progesterone, with estrogen speculated to suppress appetite and progesterone recognized for increasing appetite in the presence of estrogen (Hirschberg, 2012; McLay et al., 2007; Devries et al., 2006). Previous research has also highlighted that during the luteal phase, when both estrogen and progesterone levels rise, food intake tends to be elevated, particularly for sweet foods, while during the follicular phase, when estrogen levels rise alone, food intake decreases (Bowen & Grunberg, 1990; Racine et al., 2012, 2013; Klump et al., 2013, 2008).

The female body is highly susceptible to the influence of sex hormones, as they play a vital role in maintaining optimal reproductive function. Surprisingly, the current study discovered no significant alterations in nutritional intake throughout the menstrual cycle among the selected group of female elite athletes. These findings imply that the impact of the menstrual cycle on nutritional intake might be mitigated in female athletes due to their heightened energy requirements resulting from rigorous daily training.

Furthermore, previous research has indicated that athletes experience increased cortisol secretion due to the heightened stress associated with exercise (Schaal, Van Loan & Casazza, 2011).

The hypothalamic-pituitary-adrenal (HPA) axis is widely recognized as the primary mediator of physiological stress responses and may play a role in the connection between stress and food intake. When a perceived threat or challenge arises, corticotropin-releasing hormone (CRH) is released from the hypothalamus, triggering the subsequent release of adrenocorticotropic hormone (ACTH) from the pituitary gland, and ultimately leading to the release of glucocorticoids (GCs) from the adrenal cortex (Tsigos & Chrousos, 2002). Glucocorticoids, such as cortisol in humans and corticosterone in animals, promote glucose availability by stimulating protein breakdown, gluconeogenesis, and lipolysis, thus facilitating energy mobilization and adaptation. While cortisol is acknowledged for its contribution to increased adiposity during periods of energy surplus, it also serves as a critical catabolic hormone secreted by the adrenal cortex in response to prolonged exercise, starvation, glycogen depletion, and stress (Schaal, Van Loan & Casazza, 2011). Consequently, in athletes, cortisol secretion in response to stress may be a significant factor influencing appetite. This could potentially account for the relatively modest impact of sex hormone secretion, which fluctuates with each menstrual cycle, on appetite. In athletes, the secretion of cortisol induced by exercise can be regarded as a factor contributing to anxiety and the sustenance of elevated nutrient intake. Notably, heightened levels of psychological anxiety have been documented among female athletes with comparable profiles (Miyamoto, Hanatani & Shibuya, 2022).

The results of the present study found no significant changes in nutritional intake with the menstrual cycle in female athletes, including top-level female athletes. While the mechanisms enabling female athletes to uphold elevated nutrient intake while circumventing the influences of sex hormone production linked to the menstrual cycle remain ambiguous, we contend that this study holds noteworthy significance in its examination of nutrient intake patterns across each menstrual cycle among female athletes. Nevertheless, the underlying causal connections remain uncertain, and for a more precise comprehension of the impact of the menstrual cycle on nutrient consumption, forthcoming research should encompass assessments of sex hormone levels and pinpoint the ovulation timeframe. An inherent constraint within the study design was the inability to directly assess blood levels of female hormones in concordance with the natural menstrual cycle. We posit that a more comprehensive discourse could have ensued had these specific measurements been incorporated. Nevertheless, it is worth noting that the study’s chosen design effectively facilitated the collection of data from a diverse cohort of female athletes, encompassing individuals of elite standing. Despite these limitations, the present study provides valuable insights into the relationship between the menstrual cycle and nutritional intake in female athletes, which may guide future research in this area.

Supplemental Information

Supplemental Information 1 Raw data.

Click here for additional data file.

Additional Information and Declarations

Competing Interests

Author Contributions

Human Ethics

Data Availability

Kenichi Shibuya is an Academic Editor of PeerJ.

Mana Miyamoto conceived and designed the experiments, performed the experiments, analyzed the data, prepared figures and/or tables, authored or reviewed drafts of the article, and approved the final draft.

Kenichi Shibuya conceived and designed the experiments, performed the experiments, analyzed the data, prepared figures and/or tables, and approved the final draft.

The following information was supplied relating to ethical approvals (i.e., approving body and any reference numbers):

All participating athletes provided their consent for the study. The Ethics Committee of Niigata University of Health and Welfare provided approval to carry out the study (Approval #17982-180606).

The following information was supplied regarding data availability:

The raw data is available in the Supplemental File.

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
