# Peer review of "Exploring the relationship between nutritional intake and menstrual cycle in elite female athletes"

_PeerJ, doi:10.7717/peerj.16108_

## Round 0.1 · original submission · Minor Revisions

Thanks for your submission, please address the queries raised by the reviewers.

·

Basic reporting

Clear and unambiguous, professional English used throughout:
In the manuscript, professional English was used.


Literature references, sufficient field background/context provided:
The article provided background information about the menstrual cycle in lines 39-42 the two phases were mentioned.
Your induction needs to be more detailed for a lay audience. I suggest that you explain what happen in the follicular and luteal phases and explain the ovulation and corpus luteum. Those points can be added in line 42.
In line 49-50 the reader is informed, that estrogen has an appetite-suppressing effect and progesterone an appetizing effect, whereas it is not mentioned in which phase of the menstrual cycle estrogen and progesterone are present. Nevertheless, you do not explain In line 69-70 how you will test the hypothesis that there is no significant difference in nutrient intake during the menstrual cycle.


Professional article structure, figures, tables. Raw data shared:
The phases of the menstrual cycle were mentioned in the methods section of the abstract line 20.
In the abstract section of the article, the conclusion of the study was presented to the reader. This is one important issue for you, because the PeerJ Instructions for Authors suggest not to mention the conclusion in the abstract.
In the results section lines 109 and 132 you mentioned the tables after the authors' contribution. The next most important item is to place footnotes or a legend below the table 2. This will help to explain abbreviations, and the reader has all the necessary information below the table.
To be able to replicate the study, enough information must be present in the methods section lines 80 – 90. You have described the procedure for documenting the dietary intake during three days within a single week, but you have not mentioned if those days will be consecutive days during the week or freely chosen days within a week. This is the next important item.
The least important point is, that you did not describe why previous studies determined only total energy (kcal), macronutrients (grams and percentage), and fiber (in grams) as measuring points for nutrient intake.
All raw data was made available in table 1&2 in accordance with PeerJ Data Sharing policy, because it is saved in an accessible Excel file.


Self-contained with relevant results to hypotheses:
In the discussion section, lines 158 – 168, the effects of cortisol on energy mobilization and adjustment in the body were discussed. Athletes who have increased cortisol levels due to training and stress, the cortisol level has a significant impact on their appetite. This section is self-contained, because the reader can understand the relationship between food intake and physiological stress.
The name of the study hypothesis was formulated in the induction objective section line 68 -69 of the manuscript. The results are related to the hypothesis, because it was determined, that a smaller variation in nutrient intake relates to the menstrual cycle.
The least important point is, that the figures must be referenced. You have not mentioned, where the data was gathered for the tables, for example: (own work, data collected during survey X).

Experimental design

Original primary research within Aims and Scope of the journal:
This study, will fill the research gap by developing measures to maintain the condition of female athletes.


Rigorous investigation performed to a high technical & ethical standard:
In the study a high technical standard was used because the collected data was analysed using the lmerTest package in R. Ethical standards were met, because all athletes had to give their written informed consent prior to study enrolment. Comprehensive explanation of the study was given to the athletes. The Ethics Committee of Niigata University of Health and Welfare approved the study.


Methods described with sufficient detail & information to replicate:
To validate the parameters for the linear mixed model, the Akaike’s Information Criterion was used. The information was given, that the significance level was set as 0.05, but it was not explained what effects this has on the rejection or approval from the hypothesis. It was described, which method was used to estimate the degrees of freedom.

Validity of the findings

Impact and novelty not assessed. Meaningful replication encouraged where rationale & benefit to literature is clearly stated:
Your study is targeting a niche audience, professional female athletics. Nonetheless, your study offers added value in the literature, as the results are indicative for female athletes in terms of their nutrition during the menstrual cycle. Your study fills the research gap, which give female athletes a guidance how they can optimise their nutrition intake during the menstrual cycle. As described in the methods section lines 80 – 90 participants must record their meals, a home scale was used to measure body mass, body fat percentage. Those described tasks by participants are easy to replicate, so the conduction of a replication studies can be considered.


All underlying data have been provided; they are robust, statistically sound, & controlled:
I thank you for providing the raw data, however you have not mentioned the data repository, where your data sets are stored. Your datasets can be stored electronically or in hard copy, therefore only designated personal should have access to it. This information should be added in the following way:
i.e., data are stored securely in line with the Data Protection Act 1998 at the research facilities from University Hospitals Birmingham NHS Foundation Trust and can be accessed by the study nurse and statistician. Your data can not be used in a reproductive study, because the measuring results from each participant are not visible in table 2 nutritional intake during the menstrual cycle. Therefore, the provided data cannot be analysed with IBM SPSS 28 for comparison with a replication study.


Conclusions are well stated, linked to original research question & limited to supporting results:
There appears to be a relationship between nutritional intake and the menstrual cycle, but the study cannot evaluate exactly how this relationship works. Therefore, your research still has to determine further parameters in order to better justify this cause-and-effect principle.

·

Basic reporting

You have used clear and unambiguous, professional English throughout your results and discussion sections. Your sentences are coherent and logical, and you have used appropriate terminology and vocabulary for a scientific article.

You have provided literature references to support your findings and claims. You have also given enough background and context for your research topic and question. You have cited your sources using the defined citation schema.

Your results section follows a professional article structure, with clear subheadings for each part of your data analysis. You have presented your data using tables, and you have reported the mean, standard error, confidence interval, and statistical tests for each variable. You have also interpreted your results in relation to your hypotheses.

Experimental design

Your article is original primary research within the Aims and Scope of the journal. You have conducted a novel study on the variation of nutrient intake across the menstrual cycle among female elite athletes, which is relevant and important for the field of sports science and nutrition. You have also followed the journal’s guidelines for writing a scientific article in terms of language, structure, and references

Your research question is well-defined, relevant, and meaningful. You have clearly stated your research question and hypothesis in your introduction, and you have explained how your research fills an identified knowledge gap in the literature. You have also provided sufficient background and context for your research topic and question.

Your investigation is rigorous and performed to a high technical and ethical standard. You have recruited a large sample of female elite athletes from different sports, and you have measured their dietary intake, body mass, body composition, and menstrual cycle phases using valid and reliable methods. You have also obtained informed consent from all participants, and you have obtained ethical approval from the relevant committee. However, the consent forms are in Japanese, I couldn't confirm it.

Your methods are described with sufficient detail and information to replicate. You have provided a clear description of how you recruited your participants, how you measured their dietary intake, body mass, body composition, and menstrual cycle phases, and how you analyzed your data using linear mixed models. You have also reported the mean, standard error, confidence interval, and statistical tests for each variable.

Validity of the findings

The article does not assess the impact and novelty of the findings. However, it encourages meaningful replication of the study, emphasizing the need for measuring sex hormone levels and determining ovulation dates to further understand the impacts of the menstrual cycle on nutrient intake among female athletes.

The study provides raw data on nutritional intake, including energy, protein, fat, carbohydrate, calcium, iron, and fibre. The statistical analyses are mentioned, and significance levels are provided.

Your discussion section follows a professional article structure, with a clear summary of your main findings, a comparison with previous literature, an explanation of the possible mechanisms or implications of your results, a discussion of the limitations of your study, and a conclusion with suggestions for future research. However, you can add more to your discussion section. You may want to consider adding more to address the following points:
How do your results contribute to the existing knowledge or practice in your field?
What are the strengths and weaknesses of your study design and methods?
How do your results relate to the broader context or implications of your research topic or question? - What are the possible applications or recommendations based on your results?

The conclusions are clearly stated and linked to the original research question of examining variations in nutritional intake during the menstrual cycle among female elite athletes. The conclusions align with the findings presented in the results section and are limited to supporting the obtained results.

Additional comments

Overall, the article provides valuable insights into the relationship between nutritional intake and the menstrual cycle among female elite athletes. It meets several criteria for a well-structured research article and addresses the research question effectively. However, there are areas where additional information and data sharing could enhance the clarity, reproducibility, and validity of the study.

---

## Round 0.2 · accepted · Accept

I have reviewed your resubmitted manuscript.